# Subcritical Water Pretreatment for the Efficient Valorization of Sorghum Distillery Residue for the Biorefinery Platform

**DOI:** 10.3390/bioengineering10010038

**Published:** 2022-12-28

**Authors:** Anusuiya Singh, Chiu-Wen Chen, Anil Kumar Patel, Cheng-Di Dong, Reeta Rani Singhania

**Affiliations:** 1Department of Marine Environmental Engineering, National Kaohsiung University of Science and Technology, Kaohsiung City 81157, Taiwan; 2Sustainable Environment Research Center, National Kaohsiung University of Science and Technology, Kaohsiung City 81157, Taiwan; 3Institute of Aquatic Science and Technology, National Kaohsiung University of Science and Technology, Kaohsiung City 81157, Taiwan; 4College of Hydrosphere, National Kaohsiung University of Science and Technology, Kaohsiung City 81157, Taiwan; 5Centre for Energy and Environmental Sustainability, Lucknow 226 029, India

**Keywords:** cellulase, biomass, hydrothermal pretreatment, bioethanol

## Abstract

The depletion of fossil fuels is resulting in an increased energy crisis, which is leading the paradigm shift towards alternative energy resources to overcome the issue. Lignocellulosic biomass or agricultural residue could be utilized to produce energy fuel (bioethanol) as it can resolve the issue of energy crisis and reduce environmental pollution that occurs due to waste generation from agriculture and food industries. A huge amount of sorghum distillery residue (SDR) is produced during the Kaoliang liquor production process, which may cause environmental problems. Therefore, the SDR generated can be utilized to produce bioethanol to meet current energy demands and resolve environmental problems. Using a central composite experimental design, the SDR was subjected to hydrothermal pretreatment. The conditions selected for hydrothermal pretreatment are 155 °C, 170 °C, and 185 °C for 10, 30, and 50 min, respectively. Based on the analysis, 150 °C for 30 min conditions for SDR hydrothermal pretreatment were selected as no dehydration product (Furfural and HMF) was detected in the liquid phase. Therefore, the pretreated slurry obtained using hydrothermal pretreatment at 150 °C for 30 min was subjected to enzymatic hydrolysis at 5% solid loading and 15 FPU/gds. The saccharification yield obtained at 72 h was 75.05 ± 0.5%, and 5.33 g/L glucose concentration. This non-conventional way of enzymatic hydrolysis eliminates the separation and detoxification process, favoring the concept of an economical and easy operational strategy in terms of biorefinery.

## 1. Introduction

With an expanding population, a resource and energy crisis has arisen, requiring new technology to replace fossil fuels with renewable energy [1]. Greenhouse gas emissions are emitted by the utilization of fossil fuels for energy utilization in industry, in the domestic market, and in transportation, resulting in environmental pollution, global warming, and climate change [2,3] The worldwide energy demand could surge by 19% by 2040, as proposed by the International Energy Agency [3]. It is therefore essential to concentrate attention on issues related to the environment and energy. Because of these concerns, lignocellulosic biomass (LCB) is considered an alternative resource for renewable energies (solid, liquid, and gaseous biofuels) [3]. Sustainable bioprocessing is essential for commercial reasons, as biomass can be transformed into a wide range of products and by-products that include food, feed, materials, chemicals, and energies (fuel, power, heat). Sustainable bioprocessing technology represents an essential aspect of biorefinery [4]. Integrated biorefinery develops a broad range of value-added products with simultaneous generation of major products. Fossil-based oil refinery transformation to biomass-based biorefinery is the current focus of society. As discussed, biorefinery’s essential aspect in the modern economy is fossil-based product replacement with renewable energy sources. These are the principal objectives of a sustainable biorefinery and bioeconomy [1]. Hence, research has focused on producing bioethanol from LCB, which is omnipresent on the earth, and can later be substituted for fossil fuel to meet this planet’s energy needs [5,6,7]. LCB transformation and utilization for waste treatment, bioethanol, biofuel, and value-added product generation were accomplished simultaneously, resulting in waste valorization and a circular economy. Biofuel production globally reached 154 billion liters in 2018, of which 110 billion liters is the amount of bioethanol in the overall biofuel production, and the rest includes biodiesel [8]. In 2007, the estimated total capital investment by NREL in bioethanol plants for 2,300,000 KL bioethanol production from corn stover was USD 422.5 million [9]. Based on this, it is undeniable that biofuels (bioethanol, biomethane, biogas) and other value-added products are viable alternatives to non-renewable energy resources and reduce greenhouse gas emissions [10].

Inedible plant biomass or lignocellulosic biomass predominantly involves grasses, agricultural residues, and woody materials. The estimated annual production of lignocellulosic biomass is around 170 billion metric tons [11]. It is not easy to use and valorize LCB for biofuels and other value-added products due to the rigid, recalcitrant cell walls of plants that protect them from fungi and other pathogens [3,12,13]. Therefore, pretreatment is required to reduce biomass rigidity, recalcitrance, or physicochemical resilience to increase cellulose percentage and facilitate cellulose access to enzymes [1,14]. Pretreatment is primarily concerned with transforming the molecular structure of biomass and modifying its chemical composition. As part of the technical and economic aspects, pretreatments were used to enhance sugar production and reduce reaction times while reducing the need for chemicals and electricity [1,15]. Researchers have investigated how biomass can be pretreated before being hydrolyzed by studying various pretreatment methods. Pretreatment methods are categorized into three groups: biological, chemical, and physicochemical [16]. To increase ethanol fermentation by including pretreatments, a detailed review of the chemical additions to the pretreatment is necessary, and their relationship with microbes for the fermentation process is the most critical aspect [5]. Additionally, pretreatment produces some chemicals that act as inhibitors for enzymes during enzymatic hydrolysis. Furans, which are inhibitors, are generated from sugar hydrolysis when acidic pretreatment is applied. During mild alkaline pretreatments of biomass, phenolic acids are generated. In ionic liquid and organosolv pretreatments, the solvent can act as an inhibitor if not removed properly and can be toxic to fermentative microbes [17]. Therefore, hydrothermal pretreatment is preferred over other pretreatment techniques because it benefits the environment, is commercially profitable, and enhances hemicellulose recovery. These aspects of hydrothermal pretreatment have gained attention from all across the world [16]. In addition to being known as hot water pretreatment, hot compressed water pretreatment, autohydrolysis, or subcritical water pretreatment, hydrothermal pretreatment offers several benefits for green chemistry as a green solvent because water is used to perform the reaction at temperatures ranging from 150–220 °C and pressures ranging from 0.5 to 1.96 MPa [18]. At these critical temperatures and pressures, water begins to act like an acid [1,16,19]. The severity factor is an equation in which time and temperature interrelationships are calculated in a single variable to study the effect of changing operational conditions in hydrothermal pretreatment on biomass [19].

Once the pretreatment stage has been completed, enzymatic hydrolysis performs a crucial step in the production of ethanol. Because enzymes (cellulases) are capable of hydrolyzing lignocellulosic biomass into glucose, they have attracted considerable attention from around the world. It is possible to convert the glucose obtained from this process into ethanol. A total of three enzymes are involved in the enzymatic hydrolysis of lignocellulosic biomass. The first is endoglucanase (EC 3.2.1.4), which breaks glycosidic bonds, producing both reducing and non-reducing oligosaccharides. The exoglucanase enzyme (EC 3.2.1.91) attacks the reducing and non-reducing ends of oligosaccharides to produce cellobiose, which is then converted into glucose by the β-glucosidase (EC 3.2.1.21) enzyme [20]. Most studies have reported the importance of separating the solid-liquid phase after pretreatment to obtain solid fractions for successive bioconversion. As a consequence of solid-liquid separation, a significant amount of sugar and other soluble components present in the liquid phase are lost, thereby complicating the process and straining the wastewater treatment system [21]. In addition, the bioconversion perspective of hydrothermally pretreated liquid importance is explained [22]. It is therefore recommended to use hydrothermally pretreated whole slurry, which is a non-conventional process for the enzymatic saccharification process.

Sorghum distillery residue (SDR) is the waste generated during the Kaoliang liquor generation process. Because SDR is a Kaoliang liquor waste and is generated through a complicated procedure, 600 types of microorganisms are cultivated on wheat for four months during the brewing process. Then the sorghum and wheat utilized for microorganism cultivation are fermented together for 14 days to obtain 27 L of Kaoliang liquor, and the SDR generated during this process is around 50 kg. It is estimated that around 66.1 billion kg of SDR is generated each year around the world, and Taiwan holds only a 1% share [10]. Therefore, it is important to use SDR for the valorization process for the development of a sustainable biorefinery and bioeconomy. The main objective of this study is to comparatively analyze and evaluate the effect of operational conditions (time and temperature) SDR. The focus of this study is not only on the solid phase obtained after hydrothermal pretreatment, but also on the liquid phase that can be utilized for other value-added products. Therefore, in this study, the slurry obtained based on compositional analysis is used for enzymatic saccharification, eliminating the separation step, which adds cost to the technology when scaled up. Thus, enzymatic hydrolysis using slurry may prove to be an effective biorefinery strategy. To the best of our knowledge, no study has been reported regarding the hot water pretreatment of SDR and subjecting the slurry of SDR obtained after hot water pretreatment to the enzymatic saccharification process.

## 2. Materials and Methods

SDR was received from Kinmen Kaoliang Liquor Inc., Taiwan. Separation of SDR was performed based on the size varying from 300 µm to 500 µm. To determine cellulose, hemicellulose, and lignin contents, quantitative acid hydrolysis was performed [19].

A 0.22 µm membrane filter was used to filter samples for the quantitative analysis of glucose, xylose, arabinose, and acetic acid. HPLC with a refractive index detector at 35 °C (Agilent technology 1260 series instrument, Atlanta, GA, USA) was used, as well as a meta carb 87 H column at 60 °C, a mobile phase of 0.005 M sulphuric acid, and a flow rate of 0.65 mL/min.

### 2.1. Sorghum Distillery Residue Autohydrolysis

The SDR was milled and then subjected to hot water pretreatment in a 1:10 mass ratio of biomass to water (DI water). The pretreatment was conducted in a 100 mL closed batch reactor using an isothermal regime. An evaluation procedure was designed using operational parameters (temperature and time) that were selected for a central composite experimental design. Heat profile testing of the pretreatment was performed in terms of severity (log R0), and the severity factor was calculated according to the following equation:logR0 = [R0 Heating] + [R0 Isothermal processing] + [R0 Cooling] (1)
(2)$$\log\;R0=\left[\begin{array}{c} \int_{0}^{t_{ma}\begin{array}{c} x \end{array}}\frac{T\left(t\right)-100}{\omega } \end{array}\right]dt+\int_{ctr_{l}}^{ctr_{f}}\;exp\left[\begin{array}{c} \frac{T\left(t\right)-100}{\omega } \end{array}\right]dt+\left[\int_{0}^{t_{ma}\begin{array}{c} x \end{array}}\frac{T\left(t\right)-100}{\omega }\right]dt$$

Equation (1) explains the calculation of the severity factor as it is important to consider the temperature profiles during heating, isothermal, and cooling stages that are essentially involved in integrating the temperature vs. time profile (process intensity) [23]. In Equations (1) and (2), log R0 represents the severity factor. During the process, tma’x (min) represents the maximum time needed to achieve the target temperature, ctrl and ctrf (min) represent the time required during an isothermal cycle, T(t) (°C) represents temperature profiling regarding heating and cooling, and ω (14.75) represents the empirical parameter. The base temperature is 100 °C (it is a low-temperature value selected with no practical solubilization or depolymerization of hemicellulose) [19,23,24].

A vacuum filter was used to separate the reaction mixture into parts or phases (solid and liquid) after the pretreatment process. The solid phase obtained after the filtration process was kept in an oven at 45 °C overnight for drying purposes. The standard analytical procedure of the National Renewable Energy Laboratory (NREL) was followed for the quantitative compositional analysis of cellulose and hemicellulose. Acid hydrolysis was used to analyze biomass composition by adding 5 mL of 72% (*w/w*) sulphuric acid to 0.5 g of biomass with a moisture percentage below 20%. The reaction mixture was incubated for 1 h in a water bath at 30 °C with continuous stirring. A constant weight of 148.7 g was achieved by diluting the reaction mixture with distilled water and then autoclaving it at 121 °C for one hour [25]. By adding 4% (*w/w*) sulphuric acid to the liquid phase and autoclaving it for 30 min at 121 °C, the liquid phase was quantitatively analyzed [26]. In the hydrolysis of the liquid phase, oligomers transform into monomers. Following hydrolysis of the solid phase, the solid part was quantitatively analyzed for Klason lignin using gravimetric analysis [19]. Statistica 12 software was used for the ANOVA statistical analysis.

### 2.2. Enzymatic Saccharification

The pretreated SDR slurry (solid and liquid phase together) at 155 °C for 30 min was subjected to enzymatic hydrolysis. Enzymatic saccharification was carried out using 5% solid loading (including the whole slurry), 15 FPU/gds commercial (cellulase from *Trichoderma reseei* ATCC-26921), and 1 M citrate buffer diluted to 50 mM citrate buffer with pH 4.8. Enzymatic hydrolysis was carried out in 10 mL volumes, and deionized water was used for volume makeup. A temperature of 50 °C and a rotation speed of 150 rpm were used for enzymatic hydrolysis. For analysis, samples were taken after 0 h, 6 h, 12 h, 24 h, 48 h, and 72 h. The calculation of the yield percentage was based on Singh et al. [19] and Ruiz et al. [27].

The yield percentage calculation is as follows:(3)$$\mathrm{Saccharification}\;\mathrm{yield}\;\%\;=\;\frac{\left([\mathrm{Glucose}]+1.053[\mathrm{Cellobiose}]\right)}{1.111f[\mathrm{Biomass}]}\times 100$$

In Equation (3), [Glucose] is the glucose concentration (g/L), [Cellobiose] is the cellobiose concentration (g/L), and [Biomass] is the dry biomass concentration at the start of enzymatic saccharification (g/L). The cellulose fraction present in dry biomass is denoted by f, and 1.111 is the factor that is used in the equation to transform cellulose to equivalent glucose, and cellobiose conversion to equivalent glucose is done by the 1.053 factor [28].

### 2.3. Analytical Method

For quantitative analysis, the samples from the SDR (pretreated and untreated) and enzymatic saccharification samples obtained were filtered from a membrane filter of 0.22 µm. The samples were analyzed quantitatively for glucose, xylose, arabinose, cellobiose, and other organic acids such as acetic acid, formic acid, and lactic acid using HPLC (Agilent Technology 1260 series instrument), with a refractive index detector at 35 °C (Agilent Technology 1260 series instrument), Metacarb 87 H column with 60 °C temperature, and 0.005 M sulphuric acid mobile phase with a flow rate of 0.65 mL/min [19] was used.

## 3. Results and Discussions

### 3.1. Chemical Composition of Raw Material

SDR characterization was performed using the NREL method, and the chemical composition analyzed in untreated SDR included glucan (31.12 ± 1.21%), hemicellulose (11.84 ± 1.98%), and Klason lignin (25.89 ± 2%). In one of the previous studies, the composition of SDR in terms of cellulose (34.69%), hemicellulose (21.3%), and lignin (15.08%) percentage was reported [10]. Even though the glucan percentage is nearly similar, there is a difference in hemicellulose content as well as Klason lignin content, which is attributed to differences in the processing of biomass or batch variations. In sorghum straw waste, the glucan percentage, lignin percentage, xylan percentage, arabinan percentage, and acid-insoluble lignin percentage obtained were 32.6%, 28.5%, 2.4%, and 16.8%, respectively [29]. These studies, including the present study, suggest that SDR contains a high concentration of fermentable carbohydrates, which makes it an attractive biomass for bioethanol and other value-added products.

### 3.2. Operational Variable Effects on Solid Phase Characterization

SDR characterization was performed using the NREL method. The liquid and solid phases obtained after pretreatment were compositionally analyzed to study the effect of variables (temperature and time); the effect of variables can be seen in Figure 1. The results of pretreated SDR and the effect of severity at different temperatures (155 °C, 170 °C, 185 °C) are explained in Figure 2, Figure 3 and Figure 4. Figure 2 and Figure 3 show the effect of severity on glucan and lignin percentages at different temperatures (155 °C, 170 °C, 185 °C), respectively. Hemicellulose was not detected after 155 °C for 30 min pretreatment and is shown in Figure 4.

The highest glucan percentage obtained was 43.10% at 185 °C for 10 min. The glucan percentage at 155 °C for 50 min was 38.22 ± 1.32%, but this result is in contrast to 170 °C when the cellulose percentage was lower than that of the raw material. The reason for the lower cellulose percentage loss at low temperatures is that at 150 °C or lower temperatures, the cellulose percentage loss is low because the degree of polymerization (DP) of cellulose is higher at that temperature [30,31]. By raising the temperature above 160 °C, more cellulose becomes solubilized, resulting in increased cellulose loss [31], which explains why at 170 °C the cellulose percentage was 22.47 ± 1.43, 25.49 ± 1.33, 28.68 ± 1.72, which is lower than the cellulose percentage, i.e., 31.12 ± 1.21% obtained in the raw material. However, a reduction in cellulose loss is observed when the temperature is increased above 170 °C, as at 185 °C for 10, 30, and 50 min the cellulose percentage was 43.10 ± 2.07, 34.34 ± 0.89, and 40.28 ± 2.91, respectively. According to Ma et al. [31], the possible explanation for this could be described in two phases, which are an accelerated initial part, and a slower second part. The variation in the rate of cellulose is due to characteristic variations in cellulose reactivity. This is because the short chains of cellulose that are reactive solubilize quickly and hence cause speedy loss of cellulose. The long chain of cellulose that happens to be less reactive is accountable for the slower second part at the later phase [31]. The glucan percentage at 185 °C for 30 min (34.39 ± 0.89%) is less than the glucan percentage at 185 °C for 50 min (40.28 ± 2.91%). Glucooligosaccharide (GOS) concentration is slightly higher at 185 °C for 30 min than the GOS concentration at 185 °C for 50 min (Refer Table 1). This is because around 3.7% of glucan solubilizes or dissolves in the form of GOS obtained from β-1,3-1,4-glucan [32], and this could be the probable cause for the low glucan percentage at 185 °C for 30 min. In crystalline cellulose, the increased temperature causes intermolecular hydrogen bond cleaving. In addition, with the annealing process, Iα transforms to Iβ with better thermodynamic stability [30]. The following could be one of the other reasons for the increase in cellulose percentage at 185 °C [30]. Hydrothermal pretreatment has one disadvantage, especially when the pretreatment’s severity is intense or higher temperature, resulting in lignin’s recondensation in the solid phase obtained from hot water pretreatment [33]; this phenomenon explains the increased lignin percentage with increased severity or temperature, as at 185 °C for 50 min. The lignin percentage obtained was 69.20 ± 0.64%, which is per the phenomena reported by Batista et al. (2019). In a study reported by Ambye-Jensen et al. [34], the lignin percentage in untreated sugarcane bagasse was 16.9 ± 0.6%, and after hydrothermal pretreatment, the lignin percentage with increased temperature increased to 24.9% at 190 °C. The results obtained in the study reported by Ambye-Jensen et al. [34] are similar to the results obtained in the current study. The increased lignin percentage corresponds to the increased severity of the hydrothermal pretreatment. During hydrothermal pretreatment, the lignin dissolves, dissipates from the cell wall source, and scatters on fiber surfaces, causing the lignin fractions to reform or reorganize in lignocellulosic biomass rather than getting removed. Zhao et al. [35] found that when treated at 150 °C, hemicellulose fractions in fiberboard, chipboard, and blackboard increased by 12.6%, 13.6%, and 13.6%, respectively. However, as the temperature was increased, the hemicellulose percentage decreased considerably, and finally reached zero when the temperature reached 250 °C, which suggests that hemicellulose removal is enhanced by a gradual increase in temperature during hydrothermal pretreatment [35]. This is in agreement with the results of this analysis, because at 150 °C for 10 and 30 min, the SDR hemicellulose percentage was 4.55% and 8.59%, respectively; however, after the increase in severity, no hemicellulose percentage was detected since the raw material or untreated SDR had a lower hemicellulose percentage.

### 3.3. Variation in Liquid Phase Composition Corresponding to the Severity

At mild severity (155 °C for 30 min), log R0 = 4.26, maximum glucooligosaccharide (GOS) concentration reached 38.83 g/L), whereas at increased severity, log R0 = 4.48 (185 °C for 50 min), GOS concentration decreased to 0.93 g/L. At mild severity, glucooligomers are comparatively more likely to form as a result of the solubilization of easily reachable glucans, as glucooligomers form by hydrolyzing easily reachable glucans [36]. When crystalline cellulose is exposed to higher temperatures or increased severity, intermolecular hydrogen bonds form, resulting in more thermostable cellulose. The thermostable cellulose is less likely to solubilize in the liquid phase as GOS, which decreases GOS concentrations with an increase in severity [30]. The maximum lactic acid obtained was 9.95 g/L at 185 °C for 10 min because sugars start degrading in the presence of Lewis acid, pursuing two potential pathways [37,38]. The first pathway involves the intertransformation of glucose into fructose, resulting in levulinic acid and formic acid being generated by fructose dehydration, and the second involves triose being transformed into lactic acid [38]. With the increased severity of hydrothermal pretreatment, water forms hydronium ions that behave like Lewis acids, increasing solubilization and autohydrolysis rates. Pretreatment with liquid hot water produces a significant amount of water-soluble polysaccharides and oligosaccharides in response to increased severity factor [33,39]. In the current study, however, GOS and xylooligosaccharide (XOS) concentrations decrease with severity since the concentration of furfural and hydroxy methyl furfural (HMF) increases as well, which is the degradation compound resulting from dehydration of pentose and hexose sugar, respectively [40]. According to Alves-Ferreira et al. [36], the maximum amount of HMF generated was 1.0 g/L, with an initial glucan percentage of 3.4% [36]. In the present study, the maximum HMF concentration generated during hydrothermal pretreatment was 12.40 g/L at 185 °C for 50 min at a severity factor (4.48) corresponding to an increase in glucan degradation to HMF, and this explains how severity is responsible for the generation of degradation compounds. According to Souto et al. [41], various pretreatment conditions promote secondary product generation, such as formic, levulinic, and acetic acids [41]; for reference, see Table 1.

The formation of formic and acetic acid is stoichiometric due to the hydrolysis of hydroxymethylfurfural [42]. Formic acid and levulinic acid generation resulted due to the loss of furfural content, directly impacting cellulose amounts, because furfural can also be generated from the glucose fraction present [41]. Therefore, in the present experiment, we can observe in Table 1 that GOS and XOS production decreases with increased severity as the xylose and glucose fraction converts into secondary products, which are formic acid and lactic acid. Compared to lower severity, which is at 155 °C, the cellulose percentage in the solid phase has also not increased much at higher severity.

### 3.4. Pretreatment Slurry Subjected to Enzymatic Hydrolysis

The slurry obtained after hot water pretreatment at 155 °C for 30 min was selected because no dehydration components (furfural and HMF) were detected in the liquid phase, which interferes with the enzyme activity. The slurry obtained at 155 °C for 30 min was subjected to enzymatic saccharification. In addition, organic acids are produced by the solubilization and depolymerization of hemicellulose. In contrast to conventional saccharification methods, non-conventional enzymatic saccharification eliminates the separation of slurry into solids and liquids as well as eliminates the need for detoxification, and interconnected disposal issues, resulting in an easy and economical operation leading to increased saccharification yield [43]. In the current study, when the slurry obtained was subjected to enzymatic hydrolysis, it achieved 75.05 ± 0.5% yield percentage at 72 h, and the glucose concentration obtained during the saccharification process was 5.33 g/L; for reference, see Figure 5.

However, Oliveira et al. [43] used auto-hydrolyzed Eucalyptus globulus wood for the whole slurry experiment, and a low glucose yield was reported due to the presence of inhibitory compounds [43]. Therefore, in the present study, a low severity condition was used for the SDR’s hot water pretreatment to achieve high glucose yield during saccharification. It was reported by Balan et al. [44] in a whole slurry experiment where steam explosion pretreated biomass (whole slurry) was subjected to enzymatic saccharification and resulted in a 65% glucose yield, which is comparatively less than the yield obtained in the current study. The possible reason for the difference is the mode of pretreatment, as in the current study hot water pretreatment is used [44]. The importance of hydrolyzing pretreated slurries with a low inhibitor content is that inhibitory components can interfere with enzyme activity, resulting in a low saccharification yield [45]. However, at higher severity, inhibitor concentration increases in the liquid phase, further interfering with saccharification and fermentation [46]. Therefore, in the present study, a mild severity condition was used for biomass pretreatment to perform the enzymatic saccharification of the whole slurry for reference (see Table 1). Cellulase loading influences the enzymatic hydrolysis rate, as 18 FPU, 27 FPU, and 36 FPU per gram of sugarcane bagasse result in lower enzymatic hydrolysis rates [47]. Zheng et al. [47] reported a glucose conversion of 62.42 ± 0.82% at 9 FPU/grams of sugar cane bagasse. Saini et al. [48] reported a glucose concentration of 13.72 g/L at 72 h from pretreated pineapple leaf waste slurry when subjected to enzymatic hydrolysis at 5% solid loading [48]. The glucose concentration in the present study is lower than the glucose concentration reported by Saini et al. [48] because the Klason lignin percentage in pretreated SDR is 53.99 ± 0.5%, resulting in interference in enzyme activity. The high concentration of oligomers and phenols also sometimes contributes to interference in enzyme activity responsible for low glucose concentration. Singh et al. [19] reported a maximum yield percentage of 61.62% at 1% solid loading and 15 FPU/gds without liquid phase (washed biomass) [19], whereas in the present study, at 5% solid loading and 15 FPU/gds, the saccharification yield obtained was 75.05 ± 0.4%. This study explains the advantage of hydrothermal pretreatment at mild severity for efficient enzymatic saccharification of the whole slurry. This non-conventional method eliminates the need for a separation process, resulting in saving cost compared to the use of washed pretreated biomass.

## 4. Conclusions

SDR is generated at approximately 66.1 billion kg each year, and Taiwan holds around a 1% share. Therefore, it is essential to utilize SDR, as it is difficult to decompose such a huge amount of waste. Valorizing SDR for the production of biofuels and other value-added products through biorefineries is the most effective way to assist society and the environment. Hydrothermal pretreatment was used for the SDR pretreatment, and different parameters (time and temperature) were used for evaluation purposes. The condition that was selected after the evaluation was 155 °C for 30 min because no HMF and furfural were detected, which are degradation compounds, and a low number of organic acids produced were detected after solubilization of hemicellulose as well. It was crucial to employ slurry with little to no degradation compounds and acids for the non-conventional method of saccharification because the presence of degradation compounds and acids could negatively affect the process. The unwashed pretreated slurry of SDR (biomass + liquid phase) after hydrothermal pretreatment at 155 °C for 30 min was selected for enzymatic saccharification. The saccharification yield obtained was 75.05 ± 0.5% yield percentage at 72 h. The limitation of this process is inhibition caused by the oligomers present in the liquid phase and lignin present in the solid phase, which interfere during the hydrolysis process. Therefore, it is necessary to study the low severity effect on solid and liquid phase composition to use it (slurry) for the bioethanol production process. Hence, this non-conventional method of enzymatic hydrolysis makes a biorefinery operation simple and cost-effective because the separation process is eliminated.

## Figures and Tables

**Figure 1 bioengineering-10-00038-f001:**
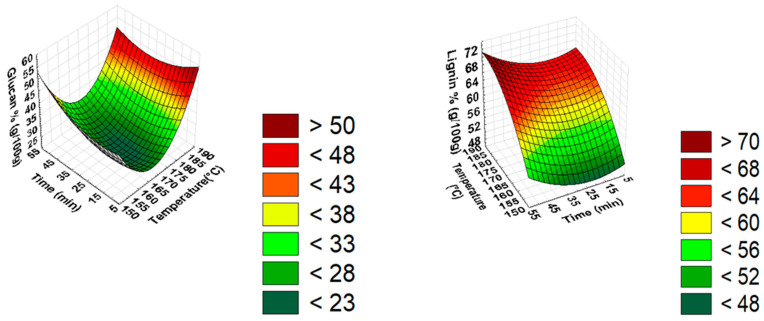
Effect of temperature and time on glucan and lignin percentage (g/100 g).

**Figure 2 bioengineering-10-00038-f002:**
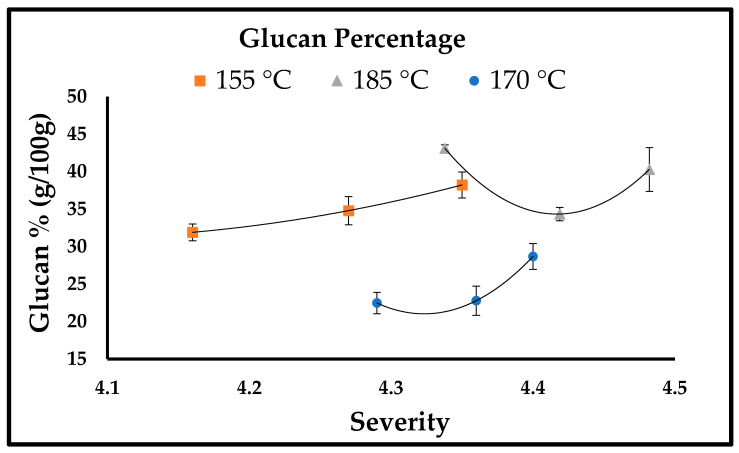
Glucan percentage (g/100 g).

**Figure 3 bioengineering-10-00038-f003:**
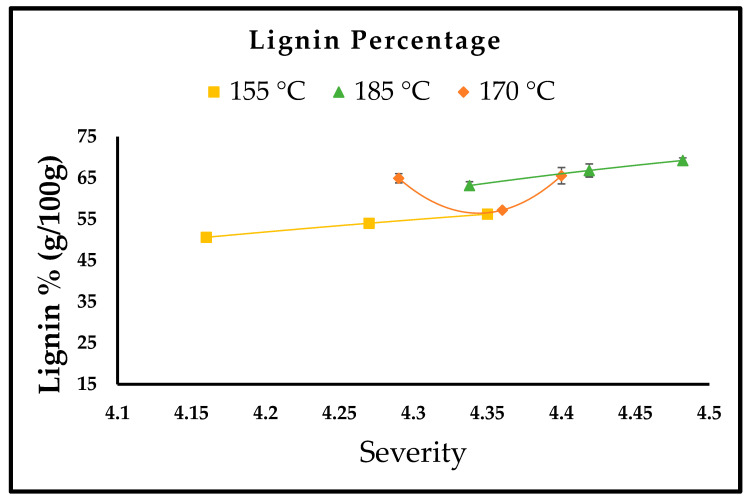
Lignin percentage (g/100 g).

**Figure 4 bioengineering-10-00038-f004:**
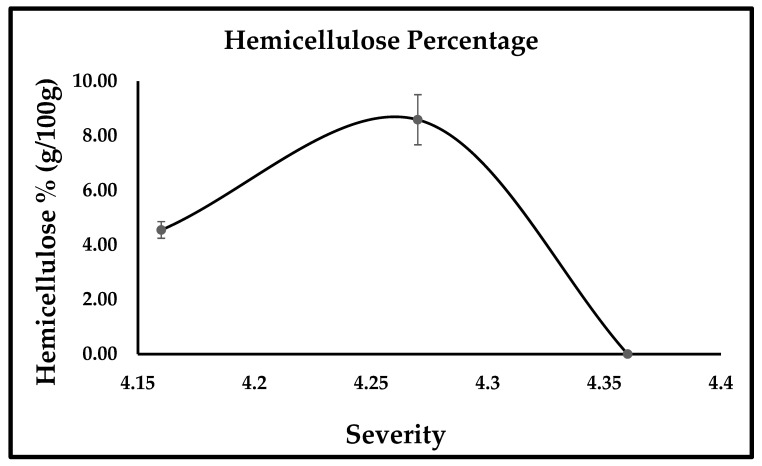
Hemicellulose percentage (g/100 g).

**Figure 5 bioengineering-10-00038-f005:**
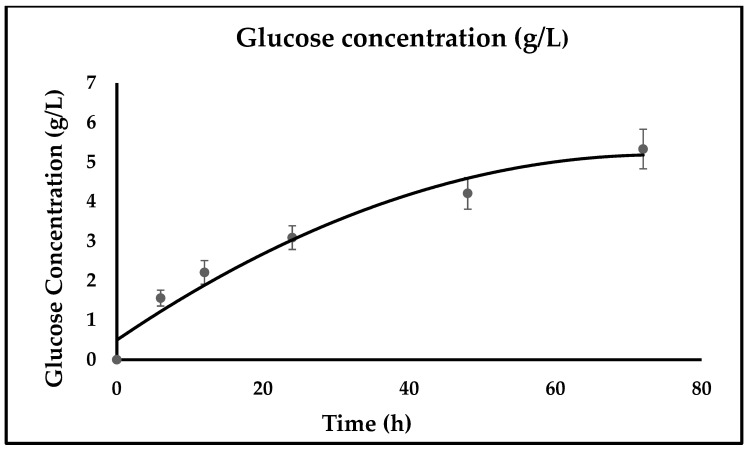
Enzymatic saccharification kinetics.

**Table 1 bioengineering-10-00038-t001:** Experimental conditions evaluated in the hydrothermal pretreatment on liquid phase of (SDR). Severity factor, pH, and heating rate.

Temperature (°C)	155 °C			170 °C			185 °C		
Time (min)	10	30	50	10	30	50	10	30	50
Log (log*R*_0_)	4.16	4.26	4.36	4.29	4.35	4.40	4.33	4.42	4.48
pH	3.55	3.55	3.54	3.62	3.57	3.57	3.61	3.43	3.40
Heating rate (°C/min)	2.09	2.07	1.99	2.00	2.05	2.07	2.06	2.03	2.01
**Liquid phase (g/L)**									
Glucose	0.0042 ± 0.00	1.82 ± 0.00	0.129 ± 0.00	1.78 ± 0.00	1.85 ± 0.07	1.42 ± 0.00	7.42 ± 0.02	5.16 ± 0.00	6.57 ± 0.28
Xylose	Nd	3.17 ± 0.06	1.90 ± 0.03	1.66 ± 0.01	1.68 ± 0.06	0.95 ± 0.01	3.65 ± 0.02	2.43 ± 0.01	Nd
Arabinose	1.52 ± 0.03	0.01 ± 0.01	1.71 ± 0.09	1.26 ± 0.01	0.81 ± 0.01	0.56 ± 0.00	1.66 ± 0.01	2.12 ± 0.01	2.12 ± 0.03
XOS	5.38 ± 0.81	4.19 ± 0.69	4.32 ± 0.11	2.48 ± 0.43	4.21 ± 0.58	1.24 ± 0.18	Nd	Nd	Nd
GOS	28.96 ± 1.81	38.83 ± 1.72	23.21 ± 1.37	17.76 ± 1.21	16.33 ± 2.53	5.91 ± 0.05	5.82 ± 0.82	4.49 ± 0.32	0.93 ± 0.74
Acetic acid	Nd	Nd	0.87 ± 0.00	0.54 ± 0.00	0.84 ± 0.00	1.02 ± 0.00	1.19 ± 0.00	2.05 ± 0.00	2.04 ± 0.00
Furfural	Nd	Nd	0.36 ± 0.00	1.43 ± 0.01	2.24 ± 0.00	1.49 ± 0.00	7.27 ± 0.03	6.98 ± 0.00	3.71 ± 0.00
HMF	Nd	Nd	1.23 ± 0.00	8.16 ± 0.03	3.54 ± 0.00	1.06 ± 0.00	8.14 ± 0.02	8.02 ± 0.00	12.40 ± 0.05
Formic acid	21 ± 0.063	0.154 ± 0.082	22.91 ± 0.48	17.01 ± 0.00	15.92 ± 0.00	13.39 ± 0.00	7.92 ± 0.00	15.66 ± 0.00	32.6 ± 0.51
Lactic acid	6.06 ± 0.01	0.05 ± 0.03	6.54 ± 0.02	4.45 ± 0.05	3.71 ± 0.02	2.54 ± 0.01	9.94 ± 0.05	6.65 ± 0.05	6.90 ± 0.01
Cellobiose	0.97 ± 0.00	0.82 ± 0.00	1.47 ± 0.00	0.11 ± 0.00	3.72 ± 0.02	0.53 ± 0.00	2.09 ± 0.03	1.16 ± 0.01	1.28 ± 0.00

XOS = xylooligosaccharides, GOS = glucooligosaccharides, Nd = not detected.

## Data Availability

Not applicable.

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
