# Peer review of "Subcritical Water Pretreatment for the Efficient Valorization of Sorghum Distillery Residue for the Biorefinery Platform"

_bioengineering, 2022, doi:10.3390/bioengineering10010038_

Round 1
Reviewer 1 Report
Please revise the text for grammatical and syntax errors. Even the title is not correct. For example, "distilleria" is not an English word. What is "biorefinery platform"? Refer to the experiments done in the past tense (not the present).
The introduction does not provide arguments that this work is worth reading. What is the novelty? What scientific gap does this work try to fill in?
Remove the guidelines from the "Results" section (This section may be divided by subheadings. It should provide a concise and precise description of the experimental results, their interpretation, as well as the experimental conclusions that can be drawn.)
Avoid writing the first letter capital in nouns when the word is within the sentence.
The increase in saccharification is the first step, but what matters is the fermentation step. There is no indication that the fermentation step would be favoured (the presence of unknown inhibitors is not impossible).
The conclusion section is too short. If this is the conclusion of this study, it makes the study rather poor.
Reviewer 2 Report
Lignocellulosic biomass or agricultural residue is a better approach to producing energy fuel (bioethanol) as it can resolve the issue of energy crisis and reduce environmental pollution that occurs due to waste generation from agriculture and food waste. A considerable amount of sorghum distillery residue (SDR) is produced during the kaoliang liquor production process, and this large amount of sorghum distillery waste generation may cause environmental problems. Therefore, the SDR generated can be utilized to produce bioethanol to meet current energy demands and environmental problems. The SDR is subjected to hydrothermal pretreatment using a central composite experimental design. Conditions selected for hydrothermal pretreatment are 155 ËšC, 170 ËšC, and 185 ËšC for 10, 30 & 50 min. Based on the analysis 150 °C for 30 minutes conditions for hydro-thermal pretreatment of SDR was selected as no dehydration product (Furfural and HMF) was detected in the liquid phase. Therefore, the pretreated slurry obtained by hydrothermal pre-treatment at 150 °C for 30 minutes was subjected to enzymatic hydrolysis at 5 % solid loading and 15 FPU/gds. The saccharification yield obtained at 72 h was 75.05 ±0.5 %, and 5.33 g/L glucose concentration. This non-conventional way of enzymatic hydrolysis eliminates the separation and detoxification process favoring the concept of an economical and easy operational strategy in terms of biorefinery. In the opinion of this referee, this work was well conducted and displayed interesting results – therefore, it must be accepted for publication after minor revision, as follow:
1. Some minor spelling mistakes were found in the manuscript – please, the text must be carefully revised.
2. “For analysis, HPLC with a refractive index detector (Agilent technology 1260 series instrument) using a meta carb 87 H column at 60°C, mobile phase (0.005 M) sulphuric acid, and a flow rate of 0.65 mL/min was used” -please, additional details concerning the used detector must be included.
3. Please, include information about the statistical approach
4. Figures 2 and 3 – Y axis must be presented using the minor value and not using “zero”.
5. Conclusion must be improved.
6. Revise carefully all cited references in order to avoid mistakes.
Reviewer 3 Report
This manuscript describes a hydrothermal pre-treatment method for lignocellulosic biomass which can enhance the production of glucose from subsequent enzymatic hydrolysis. The experimental results are interesting because progress in this area might allow for the exploitation of untapped lignocellulosic wastes. However, there are a number of areas where the manuscript can be improved.
Comment 1) The “Severity” is not defined or explained clearly and equations 1+2 need to be explained more clearly too. Since this is an important variable it would be good to explain this more clearly.
Comment 2) The abstract states “This non-conventional way of enzymatic hydrolysis eliminates the separation and detoxification process favoring the concept of an economical and easy operational strategy in terms of biorefinery”. If this is the case it would be good to mention the existing conventional approaches in your introduction and highlight how the separation methods they use would not be required if the approach you suggest is followed.
Comment 3) Your title mentions “valorization” which suggested to me that you had done some economic analysis. However, from the manuscript I could not find any economic data or information. If possible it would be good to add some representative numbers such as the potential value of bioethanol and the cost of this biomass (does it have a cost? Or do people pay for its disposal as a waste?)
Comment 4) Can bioethanol be produced directly from the output of your process? Would the by-products affect the efficiency of bioethanol production? Or are there any separation steps that would be required?
Comment 5) The conclusions are very brief and only state the main/ optimal conditions from your experiments. How could this process be improved? Or what are its limitations? It might be good to also mention the downstream processes which would be required to convert this to bioethanol.
Comment 6) A number of figures are presented with data points together with error bounds for each. However, these figures also show solid lines which I believe to be “trend lines”. It would be good to clarify the meaning of these solid lines/curves. The first time I read the manuscript I thought they might represent some theoretical model equations or some fitted correlations. If they are just “trend lines” showing the trends of the data it would be good to mention this somewhere.
Round 2
Reviewer 1 Report
This manuscript has been improved, but it is written in poor English. The authors should have turned to a professional. This remains the weak aspect of this work
Author Response
We are grateful to you for keenly observing the faults which have enabled us to improve our manuscript. We have tried our best to improve the language so that the readers can understand the authors' intentions. We hope the present manuscript may reach to your expectation.